# A Perspective on Newly Emerging Proteolysis-Targeting Strategies in Antimicrobial Drug Discovery

**DOI:** 10.3390/antibiotics11121717

**Published:** 2022-11-29

**Authors:** Janarthanan Venkatesan, Dhanashree Murugan, Loganathan Rangasamy

**Affiliations:** 1Drug Discovery Unit (DDU), Centre for Biomaterials, Cellular, and Molecular Theranostics (CBCMT), Vellore Institute of Technology (VIT), Vellore 632014, Tamil Nadu, India; 2School of Advanced Sciences (SAS), Vellore Institute of Technology (VIT), Vellore 632014, Tamil Nadu, India; 3School of Biosciences and Technology (SBST), Vellore Institute of Technology (VIT), Vellore 632014, Tamil Nadu, India

**Keywords:** PROTAC, targeted proteolysis, anti-bacterial drugs, anti-viral drugs

## Abstract

Targeted protein degradation is a new aspect in the field of drug discovery. Traditionally, developing an antibiotic includes tedious and expensive processes, such as drug screening, lead optimization, and formulation. Proteolysis-targeting chimeras (PROTACs) are new-generation drugs that use the proteolytic mechanism to selectively degrade and eliminate proteins involved in human diseases. The application of PROTACs is explored immensely in the field of cancer, and various PROTACs are in clinical trials. Thus, researchers have a profound interest in pursuing PROTAC technology as a new weapon to fight pathogenic viruses and bacteria. This review highlights the importance of antimicrobial PROTACs and other similar “PROTAC-like” techniques to degrade pathogenic target proteins (i.e., viral/bacterial proteins). These techniques can perform specific protein degradation of the pathogenic protein to avoid resistance caused by mutations or abnormal expression of the pathogenic protein. PROTAC-based antimicrobial therapeutics have the advantage of high specificity and the ability to degrade “undruggable” proteins, such as nonenzymatic and structural proteins.

## 1. Introduction

Proteolysis-targeting chimeras (PROTACs) are bifunctional protein degraders that use the E3 ubiquitin ligase pathway for the degradation of the protein of interest. A PROTAC molecule consists of three components: a ligand moiety that targets the protein of interest (POI), another ligand that binds to E3 ligase, and a linker, which bridges between these two ligands [1]. The main function of these ligands is to attract E3 ligase and POI and initiate polyubiquitination for degrading POI via the ubiquitin–proteasome system (UPS) (Figure 1) [2]. The ubiquitin–proteasome system is an essential pathway of every eukaryotic cell for maintaining homeostasis and regulating gene transcription and translation, cell cycle, and apoptosis [3]. In the ubiquitin proteolysis pathway, the ubiquitin molecule binds to the ubiquitin-activating enzyme E1. The E1-bound ubiquitin transfers the ubiquitin to the E2-conjugating enzyme, which is later transferred to E3 ligase, and, finally, ubiquitin binds to POI. These are ATP-driven cascades of reaction where the ubiquitin molecule is transferred from one molecule to another and, finally, to POI. Similarly, several ubiquitin molecules bind to the POI, which signals the proteasome to initiate the degradation of the polyubiquitinated POI. This innate protein degradation pathway is utilized for degrading POI.

The targeted degradation of the proteins has been explored using various techniques like ligand-induced degradation [4] (LID), hydrophobic tagging (HyT) [5], etc. These techniques later lead to the development of PROTACs [6]. PROTACs are more efficient than conventional small molecule inhibitors [7]. For instance, traditional small molecule inhibitors could only inhibit the activity of certain enzymes or could block the partial function of the protein, while PROTACs can completely eliminate the disease-related proteins [8]. A significantly lower concentration of the drug is required in case of targeted protein degradation using PROTACs as compared to small molecule inhibitors. Many proteins which remain undruggable over the decades, like scaffold proteins, transcriptional factors, or proteins without active binding sites, could be easily targeted by PROTACs and other similar targeted technologies [9]. Such molecules have the great advantages of high selectivity, catalytic, and drugging the undruggable targets. 

The first PROTAC molecule was successfully developed in 2001, and, to date, more than 3270 PROTACs have been developed [10]. Some of the PROTAC molecules are currently in different phases of clinical trials, and their initial results have provided a great modality for PROTAC-based degraders (Table 1) [11]. Thus, PROTACs have grabbed the attention of various pharmaceutical companies. Companies such as Arvinas [12], Pfizer [13], Accutar Biotech [14], Bristol Myers Squibb [15], Dialectic Therapeutics [16], Foghorn Therapeutics [17], Kymera Therapeutics [18], Nurix Therapeutics [19], C4 Therapeutics [20], and Cullgen [21] have already entered in the race of clinical trials for their respective PROTAC molecules. It has been predicted that within a few years, approximately 15 PROTAC molecules will be in clinics [22]. Due to the inarguable potential of PROTACs in the current era, researchers are exploring the possibilities of developing new protein degraders for various diseases, such as immunological disorders [23], inflammatory disorders [24,25,26], cancer [27,28], auto-immune diseases [29], neurological diseases [30], bacterial infections [31], and viral infections [32]. It is undeniable to state that PROTAC-based degraders are highly investigated in the field of cancer research, and many protein degraders are in the pipeline for clinical trials. However, the exploration of PROTACs in the field of anti-microbial remains marginal. This review is an attempt to highlight the state-of-the-art protein-based degraders targeting microorganisms. It also emphasizes PROTACs as an alternative to antibiotics. 

In the current scenario, due to the inappropriate/overuse of antibiotics, the use of antibiotics in the agricultural field and feeding livestock has led to the emergence of resistant strains of pathogens, which is a major threat to humankind. Consequently, pharmaceutical industries consider the development of new antibiotic as potentially effective for a shorter duration and also requires hefty investment [46]. Thus, there is a need to develop new strategies for targeting multidrug-resistant pathogens. Thus, PROTAC could be a glimmer of hope for destroying resistant pathogens [31]. Due to the characteristic nature of direct degrading of the disease-related protein or POI instead of inhibiting them could provide enhanced sensitivity towards multidrug resistant pathogens [47]. Since PROTACs utilize a chemical knockdown approach, an innate cellular mechanism, there are likely fewer chances of the generation of spontaneous mutations in the target protein [8]. There are several approaches for degrading the POI, and they are classified based on the type of degradation system used, namely, the eukaryotic system and prokaryotic system.

## 2. Eukaryotic System

### 2.1. Anti-Viral PROTACs

#### 2.1.1. Degradation of Viral Protein

In this section, PROTACs that target viral proteins have been enlisted. PROTAC-based protein degraders are highly explored in this section as compared to other strategies of protein degradation. Many viral proteins have been targeted for protein degradation (Table 2). For instance, Montrose et al. developed a peptide-based PROTAC molecule that targets X-protein, which is an essential protein required for the replication of the hepatitis B virus (HBV). It was also found that the presence of X-protein could also induce hepatocellular Carcinoma (HCC). This PROTAC molecule consists of an ODD degrons (oxygen-dependent degradation) domain, an oligomerization domain, and a cell-penetrating peptide. The ODD degrons domain binds to Von Hippel–Lindau (VHL) E3 ligase, the oligomerization domain interacts with the X-protein, and octa-arginine is used as a CPP for to ease cellular entry. In vitro studies verified the ability of peptide-based PROTACs to efficiently degrade X-protein [48]. In another study, instead of the peptide as a ligand for POI, the authors used telaprevir, an anti-viral peptidomimetic protease inhibitor. They developed three different molecules (DGY-03-081 (**2**), DGY-04-035 (**3**), and DGY-08-097 (**4**)) that target the NS3/4A protease of the hepatitis C virus (Figure 2). Lenalidomide, pomalidomide and novel tricyclic imide moiety were used as the ligands for CRBN E3 ligase. The function of NS3/4A serine protease is to cleave viral polyprotein, which acts as an essential step in viral replication [49]. Thus, degradation of NS3/4A protease via PROTAC will inhibit virion formation and multiplication. These compounds were evaluated in Hep C virus-infected HEK293T cells. Interestingly, all three degraders exhibited anti-viral activity and did not show cytotoxicity to the uninfected cells. Compound DGY-08-097 (**4**) had the highest degradation ability and the least DC_50_ value (50 nM at 4 h). One of the reasons for the increased affinity might be due to the tricyclic imide moiety in the DGY-08-097 (**4**) that showed increased affinity towards CRBN E3 ligase [50].

Similarly, the endonuclease polymerase subunit (PA) of influenza virus A was the POI for developing a novel PROTAC molecule. Asperphenalenone E (APL-16-5) (**5**) was derived from an endophytic fungus *Aspergillus,* which was used as a ligand for PA. APL-16-5 (**5**) induces degradation of the viral polymerase subunit (PA) by ubiquitin–proteasome machinery, as it can bind to both the E3 ligase enzyme (TRIM25) and PA. The endonuclease polymerase enzyme is essential in the polymerization of DNA during DNA replication. Derivatives of asperphenalenone, APL-16-1 (**6**), and APL-16-2 (**7**) were also synthesized, and the results were compared with the known anti-viral drug ribavirin. HEK293T, A549, and MDCK cells were cultured and infected with influenza virus A WSN/33 for in vitro analysis. The cytotoxicity of APL-16-5 (**5**) and APL-16-1 **(6)** against the Influenza virus was in micromolar concentration (EC_50_) 0.28 to 0.36 μM. Proteosome-mediated degradation of PA with APL-16-5 (**5**) exhibited a marked decrease in viral RNA components. Later, APL-16-5 (**5**) was evaluated against influenza virus B, hepatitis C, and Zika viruses. The results from the study confirmed that APL-16-5 is a selective inhibitor for influenza viruses. Dose-dependent studies were conducted to determine the interaction of PA with TRIM25 and concluded that compound **5** induces the destabilization of PA by ubiquitination, and thereby it degrades the PA [51].

Li et al. designed a pentacyclic triterpenoid group (PTG) containing PROTAC molecule for targeting hemagglutinin (HA) of the influenza virus. Pentacyclic triterpenoids are secondary metabolites present in various medicinal plants, and they possess significant anti-viral activity. Oleanolic acid (OA) and its derivatives are compounds that were selected as the warhead for the PROTAC molecule. OA exhibited anti-viral action against the influenza A/WSN/33 virus, and it had a moderate binding affinity with HA; thus, it became an ideal molecule for PROTAC technology. Two sets of PROTAC molecules (**8**–**10**) and (**11**–**16**) were designed and studied employing different E3 ligases, such as CRBN and VHL ligands, respectively. HEK293T cells were transfected using HA plasmids, and the level of HA degradation was studied using the synthesized PROTAC molecules. A cell viability assay, immunofluorescence microscopy assay, immunoprecipitation assay, hemagglutination inhibition assay, etc., were performed to evaluate the molecules. Compound **13** (DC_50_ = 1.44 μM) exhibited the maximum HA depletion as compared to other compounds. This was also validated by molecular docking analysis by Schrodinger Suite. Furthermore, it was concluded from these assays that the VHL ligand containing PROTACs showed better HA degradation [52].

In another independent study conducted by Xu et al., oseltamivir is an approved drug for influenza that targets influenza neuraminidase (NA). Neuraminidase is an essential enzyme for viral replication. They have used oseltamivir-based compounds for targeting neuraminidase and linked them with a discrete variety of E3 ligase ligands such as VHL or CRBN. The amino or carboxylate group of oseltamivir was modified to improve its anti-viral activity. A wide variety of linker combinations like rigid as well as flexible groups like PEG, pyridyl, triazole, and piperazinyl were also involved. A set of PROTAC combinations (**17**–**38**) were designed, and from these, N-substituted oseltamivir showed increased potency than the carboxylate-substituted compound. According to the in vitro studies, compound **27** showed the best anti-viral activity having an EC_50_ of 0.33 µM, which was almost similar to the reference drug oseltamivir phosphate (EC_50_ = 0.36 µM). Furthermore, interestingly, all the synthesized compounds do not show cytotoxicity towards the normal cells with a concentration up to CC_50_ > 50 µM. Docking studies indicated that these ternary complexes showed great hydrogen bonding and hydrophobic interactions between neuraminidase and E3 ligase [53]. From these above studies, it could be concluded that there are various strategies being evolved to target viral proteins and inhibit their replication. 

Other interesting subcategories of PROTAC are ribonuclease-targeting chimera (RIBOTAC) and nucleic acid-hydrolysis-targeting chimera (NATAC). Both strategies were used to develop novel degraders. In RIBOTAC, RNase is the degrader system, and it degrades viral RNA, while NATAC uses oligonucleotide sequences to identify the POI, and further, they could be degraded by RNase L (specific for ss-RNA). Haniff et al. developed a RIBOTAC degrader that targets the RNA genome of the SARS-CoV-2 virus (Figure 3). RIBOTAC has two major constituents- a small molecule known as C5 (**39**) and an RNA attenuator hairpin (AH). This RNA attenuator hairpin binds to the RNA genome, and C5 (**39**) recruits endonucleases present in the cell and initiates the degradation of the viral genome (Figure 4) [54]. This strategy might provide solutions for various viral infections, and the only challenge is identifying and optimizing the appropriate attenuator sequence, which could bind toward a target of interest. 

NATAC is another promising approach to PROTAC technology (Figure 5). NATAC explores the function of RNase L; RNase L is an innate RNA degrading enzyme that targets UN^N sites of the viral or single-stranded RNA. Tang and his group designed a NATAC molecule having a 5’ phosphorylated 2’-5’ polyA sequence that attracts RNase L, and another end had an antisense oligonucleotide strand targeting the spike protein of SARS-CoV-2 (Compound **40**) (Figure 4). The knockdown efficiency of spike proteins by NATAC molecules was evaluated, and it was found that NATACs could significantly reduce the RNA sequence of the spike protein. Interestingly, it was also found that RNase L could also increase the mRNA level of IFN-β and IL-6 in the host cells, thereby increasing the production of interferon and further enhancing the anti-viral response in the host cell [55].

#### 2.1.2. Degradation of Human Host Protein

In this section, PROTACs that target human host cell proteins, which are involved with viral proteins to enhance their virulence, are listed (Table 3). For instance, human cytomegalovirus (HCMV) is one of the major pathogens that cause the herpes disease. It was found that HCMV protein kinase pUL97, a viral cyclin-dependent kinase (vCDK), plays a crucial role in the generation of nuclear capsid and viral replication. During HMCV infection, viral proteins upregulate the expression of various CDK-cyclin complexes that initiate pseudomitosis, which is favorable for viral replication. Thus, CDK inhibitors targeting CDKs could be a solution for HCMV infection. In this study, THAL-SNS032, which is a protein kinase inhibitor, was used to target CDK9, and thalidomide was the E3 ligase targeting ligand. In vitro cytotoxicity studies showed that the THAL-SNS032 (EC_50_ 0.025 ± 0.001 µM) was nearly fourfold more efficient than the nonPROTAC parent compound SNS032 (EC_50_ = 0.105 ± 0.004 µM) [56]. Thus, it could be concluded that these PROTAC molecules could be a possible candidate for treating herpes disease.

The research community was doomed due to the pandemic SARS-CoV-2 and researchers across the globe were trying to provide solutions for COVID-19. This led to the discovery of a repurposing drug, i.e., indomethacin (IMN) (Compound **41**), which is an anti-inflammatory drug and was found to have anti-viral activity against the *Coronaviridae* family (Figure 6). Desantis et al. designed four PROTAC molecules (**42**–**45**) using indomethacin for targeting human prostaglandin E synthase type 2 (PGE-2). PGE-2 interacts with the NSP7 protein of SARS-CoV-2. NSF7 proteins are essential for SARS-CoV-2 replication. However, the exact mode of action of indomethacin is unknown. Furthermore, it is confirmed that degradation of PGE-2 inhibits repression of viral protein synthesis by ds-RNA-dependent protein kinase R (PKR)-mediated pathway. Since this PROTAC degrades human cellular protein, it could be classified as PROTACs targeting host protein for degradation. These four synthesized PROTAC compounds (**43**) and (**45**) showed the highest activity, with EC_50_ values of 18.1 and 21.5 μM, which was nearly five times more effective than indomethacin (EC_50_ = 94.4 μM). Molecular docking studies also described that using 6 methylene units in (**43**) and a piperazine group in (**45**) as a linker seems to be perfect for the molecule’s binding interaction with the VHL ligand. Moreover, compounds (**43**) and (**45**) exhibited anti-viral activity against β-coronavirus (i.e., HCoV-OC43), with EC_50_ values of 4.7 and 2.5 µM, respectively. These compounds were specific in that they did not show cytotoxicity against MRC-5 cells (normal uninfected cells), indicating that these PROTACs are able to specifically target infected cells. PROTACs showed effective broad-spectrum action in inhibiting two different SARS-CoV-2 strains, i.e., β-coronavirus HCoV and the α-coronavirus HCoV-229E. Due to the broad-spectrum nature of the PROTACs, these molecules could be a potential target for COVID-19 disease [57]. Y et al. developed a novel PROTAC targeting HUWE1 E3 ligase and ORF3 MERS-CoV accessory protein. ORF3 restricts apoptosis in the host cell, which could be beneficial for the virus to replicate. Hence, degradation of the ORF3 protein will induce apoptosis in the host cell, thereby restricting the spread of MERS-CoV infection. HUWE1 E3 ligase was specific towards the degradation of ORF3 since PROTACs targeting other E3 ligases such as UBR5, TRIM33, Cullin5, Cullin3, and UBR4 had no effect on the stability of ORF3. Thus, it indicates that HUWE1 E3 ligase could be a better E3 ligase for targeting ORF3 of MERS-CoV. Subsequently, MERS-CoV belongs to the same family as SARS-CoV-2; studies on SARS-CoV-2 infected cells could be another potential area of research for targeting COVID-19 [58].

## 3. Prokaryotic System

Similar to the ubiquitin-based proteolytic protein degradation system, prokaryotes also have their protein degradation system, known as the Caseinolytic protease proteolytic (ClpCP) system [59]. In Gram-positive bacteria, protein arginine phosphorylation has a crucial role in bacterial cell homeostasis [60]. Phosphoarginine-based signaling has high relevance in gram-positive bacteria like *Bacillus subtilis* and *Staphylococcus aureus*. In a protein structure, arginine side chains are phosphorylated by McsB and dephosphorylated by the YwIE enzyme. McsB and YwIE are phosphorylases and dephosphorylase enzymes of bacterium, which play a crucial role in proteolysis. This phosphorylated arginine acts as a tag or marker for the degradation of a protein. Hence, these phosphorylated arginine residues are identified by the pArg-specific reader domain, which is present in the ClpCP proteolytic complex. This leads to the degradation of the pArg-tagged protein [61]. Thus, scientists have explored bacterial proteolysis systems and developed novel PROTACs targeting the ClpCP mechanism, which are known as BacPROTACs, for degrading bacterial disease-related proteins (Table 4).

Morreale et al. synthesized four different BacPROTACs (**46–49**) using the arginine peptide linker and the targeted POI was biotin (Figure 7) [62]. ClpC protease-mediated degradation is enabled by the BacPROTAC-1 (**46**) consisting of monomeric streptavidin that targets biotin and a ClpC_NTD_ anchor which has phosphorylated arginine residues mimicking bacterial degradation tag. The characterization of the designed BacPROTAC was performed using isothermal titration calorimetry (ITC), and the results revealed that it has a high affinity towards the biotin moiety (K_D_ = 0.69 µM). The BacPROTAC-1 was further modified to obtain three more compounds, one of which (BacPROTAC-1c, **49** had nonphosphorylated arginine residue as linker, while BacPROTAC -1a and BacPROTAC-1b (**47** and **48**) had modification with the chain length spacing between the biotin and the phosphorylated arginine residue. However, these variations did not affect the degradation efficiency of these four molecules. 

Furthermore, the same group has developed new BacPROTAC molecules to target Mycobacterium tuberculosis, a pathogen that causes tuberculosis [62]. However, the phosphorylated arginine-mediated signaling pathway for protein degradation is absent in mycobacteria. Interestingly, the ClpC1P1P2 protease of mycobacteria is another similar degradation pathway that initiates degradation in the presence of phospho-guanidinium residues in the ClpC_NTD_ domain [63]. Cyclomarin A (CymA) is known to penetrate the cell membrane of mycobacteria and could be used to construct a specific protein degrader [64]. CymA-based degraders were developed by solid phase peptide synthesis, and a set of cyclic peptides (sCym-1) were synthesized to target the bromodomain 1 (BD-1). JQ1 is a small molecule inhibitor that binds to BD-1, which is used as one of the ligands to target the BD-1 protein. BacPROTAC-2 (compound **50**) was made by linking the sCym-1 with the biotin moiety. Another set of BacPROTACs, BacPROTAC-3 and 3_a_ (**51** and **52**), were made by linking the JQ1 ligand with the stereochemical modifications, i.e., S and R forms, respectively. BacPROTAC-4 and 4_a_ (**53** and **54**) were made with the diastereomers of dCymM linked with the JQ1(S) ligand. BacPROTAC-5 and 5_a_ (**55** and **56**) were constructed using a different linker group, connecting dCymM with JQ1 ligands in S and R forms, respectively. These synthesized BacPROTACs showed the efficient degradation of the targeted protein in the mycobacteria, and compound **50** showed the highest binding affinity. Furthermore, compounds **51** and **55** showed only partial degradation towards the targeted protein ever at higher concentrations [62]. Thus, it could be concluded that compound **50** could be a potential target for curbing tuberculosis; however, further preclinical studies are essential to evaluate its effectiveness. Junk et al. developed a similar BacPROTAC approach towards the mycobacteria by inducing the ClpC1-mediated degradation. A simplified desoxycyclomarin derivative was used as the active compound for the development of BacPROTAC against the tuberculosis H37Rv strains residing on the THP-1 macrophages. Homo-BacPROTAC molecule was made by dimerizing the cyclomarin molecules, and it was known to show high degradation efficiency at the sub-nM level [65].

**Table 4 antibiotics-11-01717-t004:** Comprehensive information on targeted degradation of bacterial protein in the prokaryotic system using PROTAC.

S. No.	Pathogen	Protein of Interest (POI)	POI Ligand	E3 Ligase Ligand	Research Outcome	Ref.
1	*Bacillus subtilis*	ClpCP protease	Phosphorylated arginine residue	Biotin	Incubation with 100 µM of BacPROTAC showed efficient degradation The nonphosphorylated arginine linkage doesn’t show the degradation of the model protein.	[62]
2	*Mycobacterium tuberculosis*	ClpC1P1P2 protease	Cyclomarin A (CymA)	Biotin JQ1(R) and JQ1(S)	JQ1(S) BacPROTAC compound 46 showed the maximum degradation efficiency	[64]
3	*Mycobacterium tuberculosis*	ClpC1-NTD protease	Desoxycyclomarin derivative	-	Homo-BacPROTAC molecule made by dimerizing the cyclomarin molecules was known to show high degradation efficiency at the nM level.	[65]

Another interesting area of research is the utilization of the anti-tuberculosis antibiotic pyrazinamide as a degradation tag of its bacterial target. Pyrazinamide pro-drug converted into pyrazinoic acid by pyrazinamidase, an enzyme found in M. tuberculosis. It has been observed that pyrazinoic acid binds to PanD, aspartate decarboxylase, an essential and unique bacterial enzyme. These bindings induce conformational changes, which results in the degradation of PanD via the ClpP1P2 degradation pathway. Thus, this idea could be used to design novel bacterial protein degraders [66]. Long et al. developed a PROTAC-based strategy for protein degradation using tert-butyl carbamate-protected arginine. It was discovered that inhibitors linked to Boc_3_Arg could effectively degrade the inhibitor-bound protein. Thus, using this strategy, the degradation of GST-α1 (glutathione) in cancer cells was proved. Similarly, Boc_3_Arg-bound trimethoprim (TMP) showed reducing of *Escherichia coli* dihydrofolate reductase (eDGFR). Trimethoprim is an inhibitor of DGFR. It was observed that the degradation process is rapid, that within 5 h of administration, there is a decrease in the progress level by 30–80% [67]. Thus, it could be concluded that by harnessing the power of bacterial proteolytic systems, various PROTAC molecules have been created. These novel protein degraders have the potential to change the future in the field of antibiotics.

## 4. Patent Analysis and Future Perspectives

Su, Xiangdong, et al. invented and patented an anti-viral PROTAC molecule having the general formula LGP-LK-LGE, wherein LGP is a ligand for binding a deoxyribonucleic acid (DNA) polymerase; LGE is a ligand for binding an E3 ubiquitin ligase, and LK is a bridging chain linking the two above ligands. These compounds degrade deoxyribonucleic acid (DNA) polymerase and hence inhibit virus replication and kill viruses such as hepatitis B and HIV [68]. Haibing et al. made a novel oseltamivir-based PROTACs and filed a patent, which can degrade influenza virus neuraminidase and, thus, display the activity of inhibiting the replication of the influenza virus [69].

Although anti-viral PROTACs or bacPROTACs have the potential to become therapeutic agents, efficient delivery methodologies are required. Proteolysis-targeting chimeras (PROTACs) must be cell permeable to reach their target proteins, especially to cross the dense bacterial cell wall. The incorporation of PROTAC-like molecules with nanoparticles will precisely take these degrader molecules to the diseased site, where the off-target effects and the lower cell permeability could be greatly minimized [70]. Though there are 600 E3 ligases encoded in the human genome, only 4 of them are targeted in PROTAC-based degradation. Thus, the exploration of ligands for other E3 ligases might improve the efficacy of PROTACs [71]. Current knowledge of protein the degradation mechanism in pathogens is very limited compared with oncology applications; hence, a comprehensive understanding of the E3 ligases expressed in the host cell as well as in the infectious pathogen is required to design effective anti-infective PROTACs. Combining the PROTAC technology with some other therapeutic techniques, such as photodynamic therapy (PDT), could open a different way in the targeted protein degradation in a more efficient, cost-effective, and controlled manner [72]. The field of targeted protein degradation in bacteria and viruses will grow in the next future, leading to the development of chemically induced knockdowns of disease-causing proteins. Finally, anti-viral PROTACs or bacPROTACs will open a plethora of opportunities to develop next-generation antimicrobial therapies.

## 5. Conclusions

The emergence of many drug-resistant, mutant strains of pathogens has become a serious challenge to human health. It creates a need for anti-microbial drugs that follow a novel mode of action. In the current scenario, the efficacy of antibiotics is highly compromised due to the mutation in the microbial genome and the emergence of drug-resistance micro-organisms. Targeted protein degradation is a technology that could specifically degrade disease-related proteins, such as endotoxins, surface protein markers, or even viral RNA using the ubiquitin–proteasome system (UPS) and/or prokaryotic proteolysis system. Both pathways are the innate system that could eliminate these disease-related proteins. Targeted protein degradation for the elimination of proteins by hijacking the cell’s own innate mechanism is a promising technique in the case of oncogenic proteins. The same approach has been extended for the development of anti-microbials. Targeted protein degradation using PROTACs, RIBOTAC, and NATACs for the development of anti-microbial drugs could be the next-generation solution for microbial infections. These techniques are capable of degradation of infectious pathogen-derived proteins. Thus, it could be predicted that PROTACs could be a futuristic solution for microbial infections.

## Figures and Tables

**Figure 1 antibiotics-11-01717-f001:**
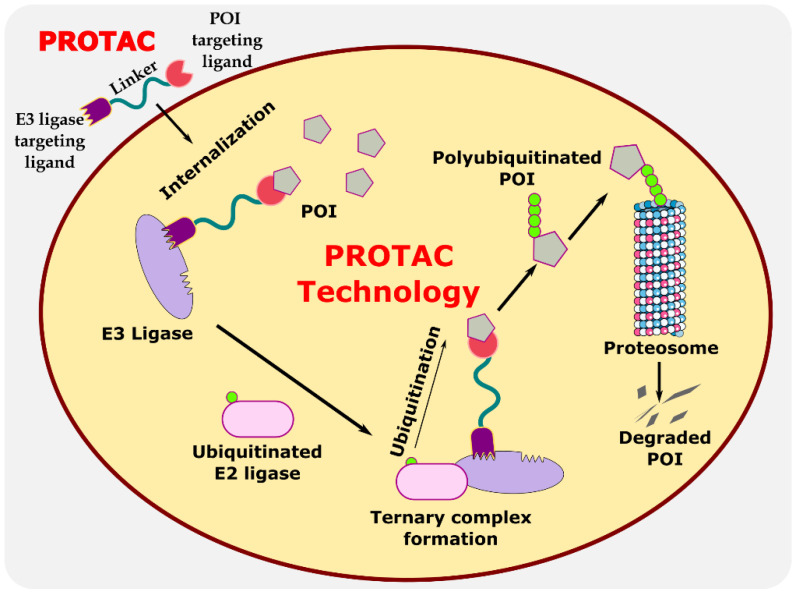
Illustration explaining the mechanism of PROTAC in targeted protein degradation.

**Figure 2 antibiotics-11-01717-f002:**
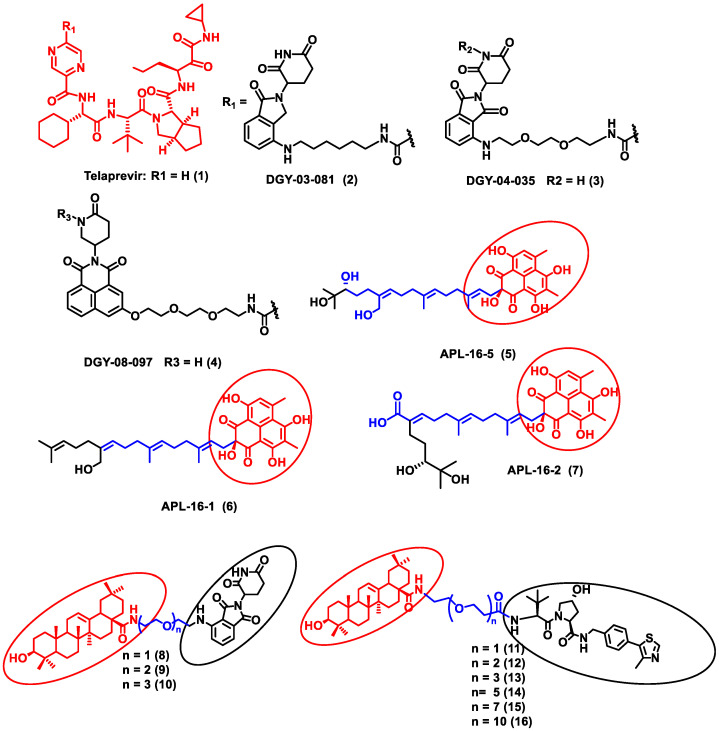
Structures of the PROTAC molecules used for the degradation of viral proteins. The red circle indicates the POI ligand. The blue wavy line indicates the linker, and the black circles indicate the E3 ligand moiety.

**Figure 3 antibiotics-11-01717-f003:**
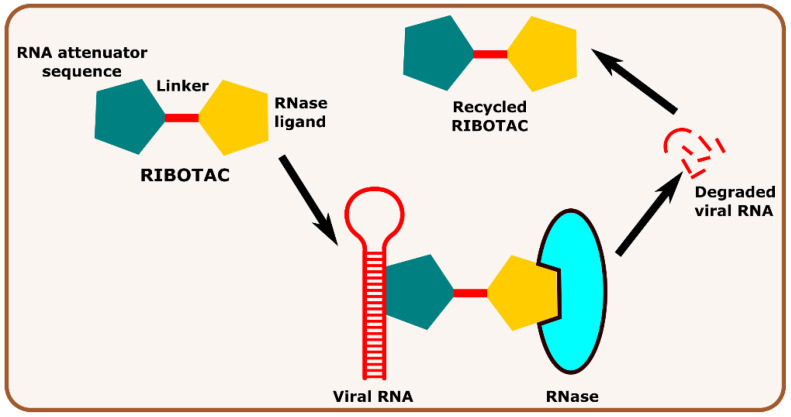
Schematic representation of the mechanism of action of RIBOTAC molecules in targeting viral RNA.

**Figure 4 antibiotics-11-01717-f004:**
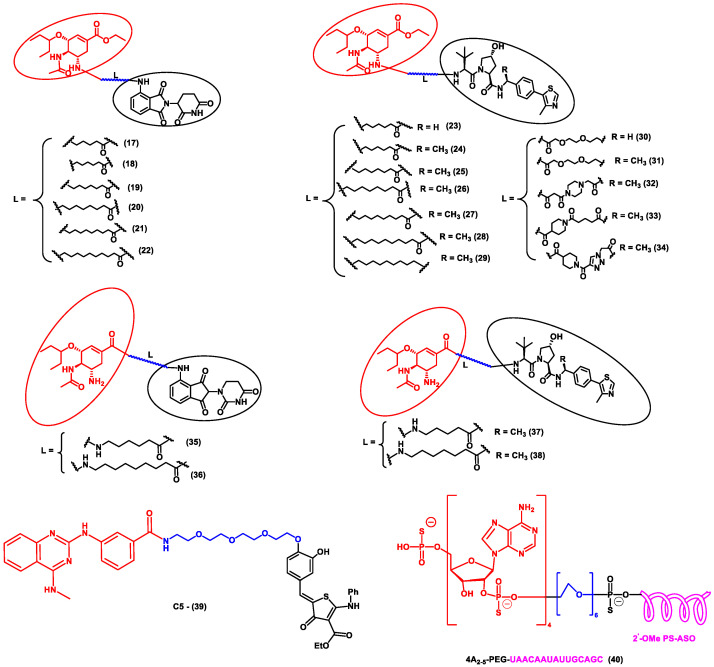
Structures of the PROTAC molecules are used for the degradation of viral proteins. The red circle indicates the POI ligand; the blue wavy line indicates the linker; the black circles indicate the E3 ligand moiety.

**Figure 5 antibiotics-11-01717-f005:**
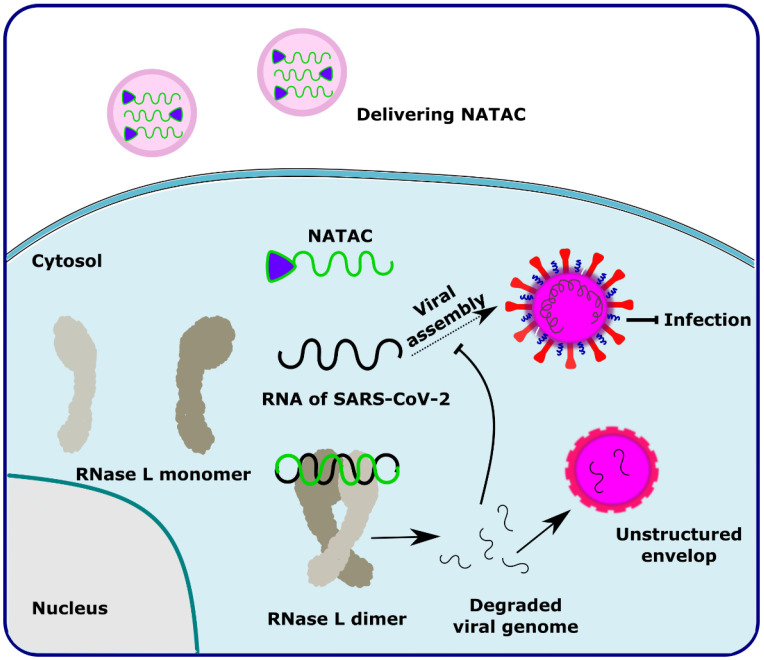
Schematic representation of the mechanism of action of NATAC molecules in targeting the SARS-CoV-2 virus. Adapted from [55].

**Figure 6 antibiotics-11-01717-f006:**
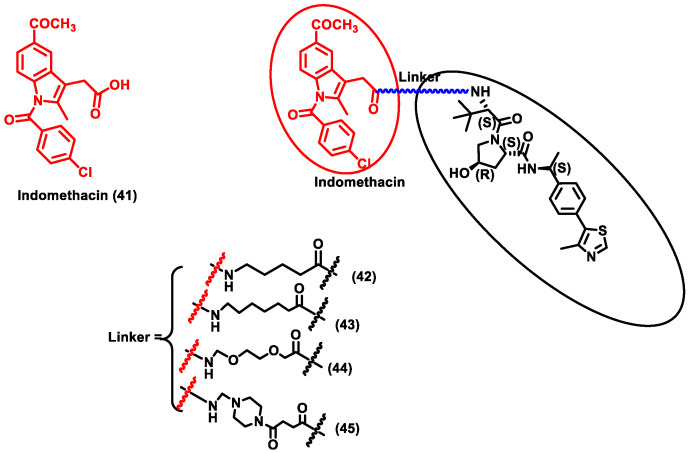
Structures of the PROTAC molecules are used for the degradation of host proteins. The red circle indicates the POI ligand; the blue wavy line indicates the linker; the black circle indicates the E3 ligand moiety.

**Figure 7 antibiotics-11-01717-f007:**
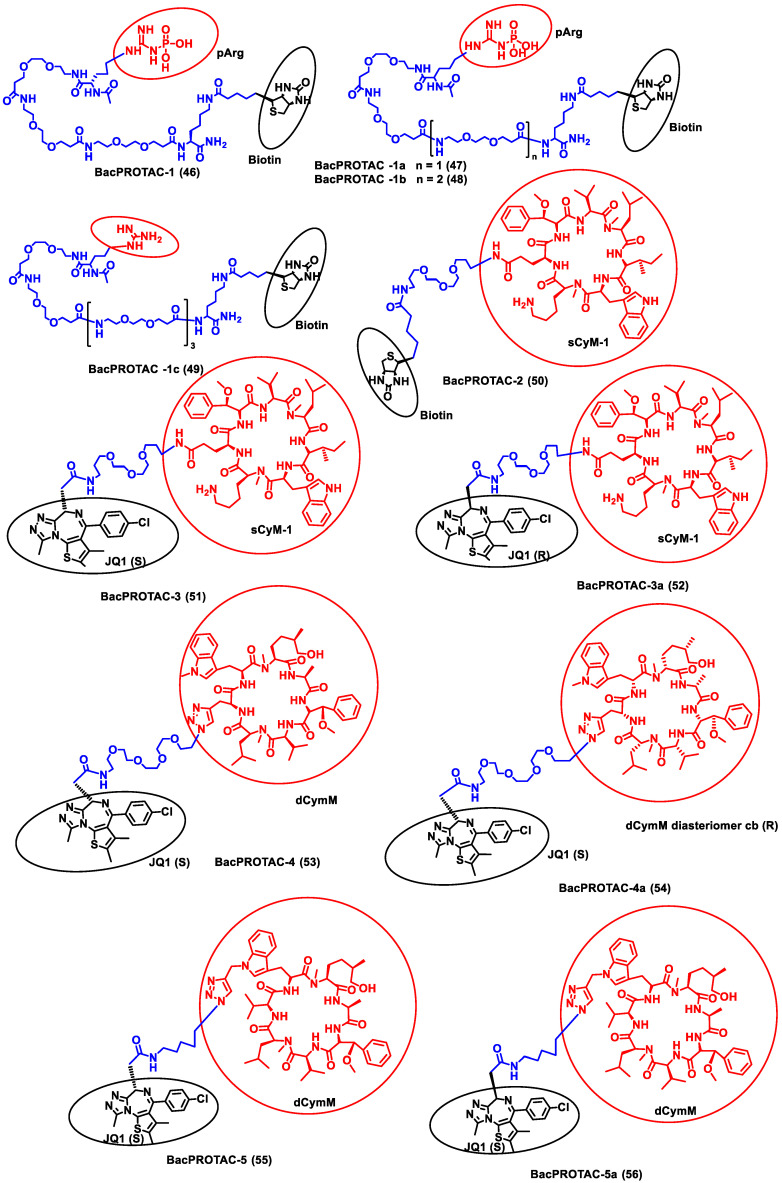
Structures of the PROTAC molecules are used for the degradation of proteins in prokaryotic cells using a bacterial proteolysis system. The red circle indicates the POI ligand. The Blue wavy line indicates the linker, and the black circle indicates the E3 ligand moiety.

**Table 1 antibiotics-11-01717-t001:** Comprehensive representation of clinical cases of PROTACs.

Sr. No.	Molecule	Route of Delivery (Dose)	Stage of the Trial	No. of Patients	Targeted Disease	Company	Follow Up Period	Clinical Trial No.	Ref.
1	ARV-110	Oral (Tablets) once or twice daily for 28 day cycles	Recruiting (Phase II)	36	Prostate cancer	Arvinas, USA	28 days	NCT03888612	[33]
2	ARV-471	Oral	Recruiting (Phase II)	36	Breast cancer	Arvinas, Pfizer, USA	28 days	NCT04072952	[34]
3	AC682	Oral	Recruiting (Phase I)	42	Breast cancer	Accutar Biotech, USA	18 months	NCT05080842	[35]
4	ARV-766	Oral	Recruiting (Phase II)	60	Prostate cancer	Arvinas, USA	6 weeks	NCT05067140	[36]
5	CC-94676	Oral	Recruiting (Phase I)	40	Prostate cancer	Celgene, USA	-	NCT04428788	[37]
6	DT2216	Intravenous administration	Recruiting (Phase I)	24	Liquid and solid tumors	Dialectic Therapeutics, USA	28 days	NCT04886622	[38]
7	FHD-609	Intravenous administration	Recruiting (Phase I)	70	Synovial sarcoma	Foghorn Therapeutics, USA	6 weeks	NCT04965753	[39]
8	KT-474	Oral	Completed (Phase I)	124	Autoimmune diseases (e.g., AD, HS, RA)	Kymera Therapeutics, USA	28 days	NCT04772885	[40]
9	KT-413	Intravenous administration	Recruiting (Phase I)	80	Diffuse large B cell lymphoma (MYD88-mutant)	Kymera Therapeutics, USA	18 months	NCT05233033	[41]
10	KT-333	Intravenous administration	Recruiting (Phase I)	80	Liquid and solid tumors	Kymera Therapeutics	18 months	NCT05225584	[41]
11	NX-2127	Oral	Recruiting (Phase I)	130	B cell malignancies	Nurix Therapeutics, USA	6 months	NCT04830137	[42]
12	NX-5948	Oral	Recruiting (Phase I)	130	B cell malignancies and autoimmune diseases	Nurix Therapeutics, USA	100 days	NCT05131022	[43]
13	CFT8634	Oral	Recruiting (Phase II)	110	Synovial sarcoma	C4 Therapeutics	90 days	NCT05355753	[44]
14	Protac MyFit	Intravenous administration	Recruiting (Phase I)	240	Sensory impairment (SPD)	University of Southern Denmark	21 days	NCT04173871	[45]

**Table 2 antibiotics-11-01717-t002:** Comprehensive information on the targeted degradation of viral protein in the eukaryotic system using PROTAC molecules.

S. No.	Pathogen	Protein of Interest (POI)	POI Ligand	E3 Ligase Ligand	Research Outcome	Ref.
1	Hepatitis B virus (HBV)	X-protein of the hepatitis B virus (HBV)	Oxygen-dependent degradation (ODD) domain of hypoxia-inducible factor (HIF-1a)	VHL ligand	Peptide-based PROTACs–efficient degradation of X-protein	[48]
2	Hepatitis C virus (HCV)	HCV NS3/4A protease	Telaprevir	CRBN ligand	DGY-08-097 more efficient than DGY-03-081, DGY-04-035	[50]
3	Influenza A virus	Viral endonuclease PA	Asperphenalenone E		APL-16-5 (Compound 5) exhibited selective degradation towards influenza virus (EC_50_ = 0.28 μM)	[51]
4	Influenza A Virus	Influenza hemagglutinin	Pentacyclic triterpenoid	VHL and CRBN ligand	Compound 13 has a longer linker and exhibited efficient degradation. (DC_50_ = 1.44 μM)	[52]
5	H1N1 influenza virus	Neuraminidase	Oseltamivir	VHL and CRBN ligand	The best anti-viral activity was seen in compound 27 (EC_50_ = 0.33 µM)	[53]
6	SARS-CoV-2 virus	The RNA genome of the SARS-CoV-2 virus	RNA attenuator hairpin (AH)		Degradation of the viral RNA genome	[54]
7	SARS-CoV-2 virus	Spike protein of SARS-CoV-2	Antisense oligonucleotide		Significantly reduced the RNA sequence of the spike protein Increase the mRNA level of IFN-β and IL-6 in the host cells and hence interferon production for anti-viral action	[55]

**Table 3 antibiotics-11-01717-t003:** Comprehensive information on the targeted degradation of human host protein using PROTACs.

S. No.	Pathogen	Protein of Interest (POI)	POI Ligand	E3 Ligase Ligand	Research Outcome	Ref.
1	Human cytomegalovirus (HCMV)	Anti-HCMV AD169-GFP activity	Cyclin-dependent kinase (CDK) inhibitor SNSO32	CRBN ligand	THAL-SNS032 exhibited EC_50_ = 0.025 ± 0.001 µM, which is nearly fourfold more efficient than the nonPROTAC parent compound SNS032 (EC_50_ = 0.105 ± 0.004 µM)	[56]
2	Syndrome coronavirus-2 (SARS-CoV-2).	Human prostaglandin E synthase type 2 (PGES-2)	Indomethacin	VHL ligand	Compounds with 6 methylene group and piperazine as linkers-great efficiency. (EC50 = 4.7 and 2.5 μM) respectively	[57]

## Data Availability

Not applicable.

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
