# Peer review of "A Perspective on Newly Emerging Proteolysis-Targeting Strategies in Antimicrobial Drug Discovery"

_antibiotics, 2022, doi:10.3390/antibiotics11121717_

Round 1
Reviewer 1 Report
In the current review article entitled “A perspective on newly emerging proteolysis-targeting strategies in antimicrobial drug discovery” Loganathan and colleagues demonstrated the current novel modalities of protein degradation approaches toward antimicrobial drug discovery. PROTACs are hetero bifunctional molecules, which induce close proximity of both E3 ligase and the target protein and cause poly ubiquitination followed by its degradation. The PROTACs have revolutionized the current drug discovery efforts by complete elimination of the aberrantly expressed proteins rather than temporal inhibition and further demonstrated to work in a catalytic manner.
Although many reviews appeared in the field of PROTACs, the current perspective mainly focuses on targeting pathogenetic proteins. At this point, I should appreciate the authors' efforts in bringing the collective information to the readers.
In my opinion, this review article has enough novelty to meet the requirements for publication in Antibiotics after addressing the following comments:
1. Can authors please cite " A. Hershko and A. Ciechanover, Annu. Rev. Biochem., 1998, 67, 425-479"
2. Can authors change the E2 ligase to E2-conjugating enzyme on page 1?
3. Can authors modify it to “Such molecules have a great advantage of high selectivity, catalytic, and drugging the undruggable targets” on page 2?
4. On page 5, please replace the oligonucleotide domain with the oligomerization domain.
5. On page 7 - line 16, does it compound 9 or 5?
6. On page 8 – line 8, CNBN should be changed to CRBN.
7. The authors stated that the VHL ligand containing PROTACs showed better HA degradation, but compound 10, which is CRBN based showed maximum HA depletion. The authors should correct this.
8. The authors should replace Figure 4, as it was copied from the original paper.
9. Please include the NATAC structure in Figure 5.
10. The authors should replace the sentence “Another research group from China” in a professional way.
11. The chemical structures of compounds 51 and 52 are identical, please correct them.
In addition, the authors should correct the typo mistakes and figure numbers throughout the article.
Author Response
Response to Reviewers
Reviewer 1
In the current review article entitled “A perspective on newly emerging proteolysis-targeting strategies in antimicrobial drug discovery”. Loganathan and colleagues demonstrated the current novel modalities of protein degradation approaches toward antimicrobial drug discovery. PROTACs are hetero bifunctional molecules, which induce close proximity of both E3 ligase and the target protein and cause poly ubiquitination followed by its degradation. The PROTACs have revolutionized the current drug discovery efforts by complete elimination of the aberrantly expressed proteins rather than temporal inhibition and further demonstrated to work in a catalytic manner.
Although many reviews appeared in the field of PROTACs, the current perspective mainly focuses on targeting pathogenetic proteins. At this point, I should appreciate the authors' efforts in bringing the collective information to the readers.
In my opinion, this review article has enough novelty to meet the requirements for publication in Antibiotics after addressing the following comments:
Answer: We thank the Reviewer for appreciating our work and for her/his encouraging comments on our review article.
- Can authors please cite " A. Hershko and A. Ciechanover, Annu. Rev. Biochem., 1998, 67, 425-479"
Answer: We thank the reviewer for this suggestion, we have included the above-mentioned reference (Ref no- 37) (Page no-1, line no-37).
- Can authors change the E2 ligase to E2-conjugating enzyme on page 1?
Answer: As suggested by the learned reviewer, we have changed E2 ligase to an E2-conjugating enzyme (Page no-1, line no-39).
- Can authors modify it to “Such molecules have a great advantage of high selectivity, catalytic, and drugging the undruggable targets” on page 2?
Answer: We thank the reviewer for his/her valuable suggestion, we have changed the statement as suggested (Page no-2, lines no-61 to 62).
- On page 5, please replace the oligonucleotide domain with the oligomerization domain.
Answer: We thank the reviewer for this suggestion, we have changed the oligonucleotide domain with the oligomerization domain. (Page no-5, line no-107, 108).
- On page 7 - line 16, does it compound 9 or 5?
Answer: We thank the reviewer for this suggestion, it is compound 5, and we have corrected them (Page no-7, line no-141).
- On page 8 – line 8, CNBN should be changed to CRBN.
Answer: We thank the reviewer for this suggestion, and we have rectified the error. (Page no-8, line no-156).
- The authors stated that the VHL ligand containing PROTACs showed better HA degradation, but compound 10, which is CRBN based showed maximum HA depletion. The authors should correct this.
Answer: We thank the reviewer for this suggestion, and we have corrected the error. (Page no-8, line no-160).
- The authors should replace Figure 4, as it was copied from the original paper.
Answer: As per the suggestion from the learned reviewer we have redrawn a similar diagram explaining the mechanism of NATAC (Page no-9, Figure 4).
- Please include the NATAC structure in Figure 5.
Answer: We thank the reviewer for this suggestion, and we have included the structure of NATAC structure (Compound 40) (Page no-10, Figure 5).
- The authors should replace the sentence “Another research group from China” in a professional way.
Answer: We thank the reviewer for this suggestion, and we have replaced “Another research group from China” with “Y et al.” (Page no-12, lines no- 254-255).
- The chemical structures of compounds 51 and 52 are identical, please correct them.
Answer: We thank the reviewer for their suggestion, and we had missed the stereochemistry, and we have corrected them. We missed giving unique numbers to a few compounds. Now we corrected all the compound structures with proper numbers, and the same has been highlighted. The numbers 51 and 52 have been changed now to 55 and 56 (Page no-17, figure 6 compounds 55 and 56).
- In addition, the authors should correct the typo mistakes and figure numbers throughout the
article.
Answer: We thank the reviewer for their suggestion, and we thoroughly checked the entire manuscript and corrected the typo errors.
Reviewer 2 Report
The review submitted by Venkatesan, et al. summarized the recent advances of antimicrobial PROTACs and other similar ‘PROTAC-like’ techniques to degrade pathogenic target proteins. The review article is well written and organized, providing a deep insight in the development of antimicrobial PROTACs. All the figures and tables are prepared and presented properly. In short, this is an informative and instructive review worth publication.
Some minor concerns:
1. Page 2, Line 62, “and to date, 3270 of PROTACs have been developed”, this sentence is not accurate since the amount of PROTACs is always changing. It is better to be “and to date, more than 3270 of PROTACs have been developed”
2. Page 8, Line 174, the citation of Figure 3 is irrelevant to the text.
3. The “E3 ligase” in all Tables should be presented consistently. For example, in Table 3, there is “Thalidomide” in line 1, does it should be described as CRBN ligand? In addition, the reviewer considers that the column “E3 ligase” should be corrected as “E3 ligase ligand”.
4. In Table 2, “pathogen” should be “Pathogen”; In Table 4, “Sr.No.” should be “S.No.”
5. The references have some problems, for example, Ref.69, the link is not accessible; Ref.70, the authors’ names are in Chinese; Ref.73 is not accessible for the reviewer (may also be not accessible for the readers). Please correct or replace these references.
6. The Abbreviations list is incomplete, please carefully check throughout the whole manuscript.
Author Response
Response to Reviewers
Reviewer 2
The review submitted by Venkatesan, et al. summarized the recent advances of antimicrobial PROTACs and other similar ‘PROTAC-like’ techniques to degrade pathogenic target proteins. The review article is well written and organized, providing a deep insight in the development of antimicrobial PROTACs. All the figures and tables are prepared and presented properly. In short, this is an informative and instructive review worth publication.
Answer: We thank the Reviewer for appreciating our work and for her/his encouraging comments on our review article.
Some minor concerns:
- Page 2, Line 62, “and to date, 3270 of PROTACs have been developed”, this sentence is not accurate since the amount of PROTACs is always changing. It is better to be “and to date, more than 3270 of PROTACs have been developed”
Answer: We thank the reviewer for their valuable suggestion, and we have changed the sentence (Page no-2, lines 63-64)
- Page 8, Line 174, the citation of Figure 3 is irrelevant to the text.
Answer: We thank the reviewer for this suggestion, we have corrected the citation.
- The “E3 ligase” in all Tables should be presented consistently. For example, in Table 3, there is “Thalidomide” in line 1, does it should be described as CRBN ligand? In addition, the reviewer considers that the column “E3 ligase” should be corrected as “E3 ligase ligand”.
Answer: We thank the reviewer for this suggestion, and we have corrected the error. (Page no- 6, Table no-2) (Page no-11, Table no-3) (Page no-15, Table no-4).
- In Table 2, “pathogen” should be “Pathogen”; In Table 4, “Sr.No.” should be “S.No.”
Answer: We thank the reviewer for this suggestion, and we have changed the “pathogen” to “Pathogen” (Page no- 6, Table no-2), and in Table 4, “Sr.No” has been changed to “S. No” (Page no-15, Table 4).
- The references have some problems, for example, Ref.69, the link is not accessible; Ref.70, the authors’ names are in Chinese; Ref.73 is not accessible for the reviewer (may also be not accessible for the readers). Please correct or replace these references.
Answer: As suggested by the learned Reviewer, we have rechecked these references and updated them. Due to the addition of one reference (suggested by Reviewer 1), the reference numbers have been changed to Ref-70, 71, and 74.
- The Abbreviations list is incomplete, please carefully check throughout the whole manuscript.
Answer: We thank the reviewer for his/ her valuable suggestion, and we have added new abbreviations (Page no- 20, Line no-405, 409, 416, 417, 419,422, 423, 428-432, 436, 437).